# Determination of the Waterway Parameters as a Component of Safety Management System

Andrzej Bąk [1,*] and Paweł Zalewski [1]

Faculty of Navigation, Maritime University of Szczecin, Wały Chrobrego St. 1-2, 70-500 Szczecin, Poland; p.zalewski@am.szczecin.pl
* Correspondence: a.bak@am.szczecin.pl

**Abstract:** This article presents the use of a computer application codenamed "NEPTUN" to ascertain the waterway parameters of the modernised Świnoujście–Szczecin waterway. The designed program calculates the individual risks in selected sections of the fairway depending on the input data, including the parameters of the ship, available water area, and positioning methods. The collected data used for analyses in individual modules are stored in a SQL server of shared access. Vector electronic navigation charts of S-57 standard specification are used as the cartographic background. The width of the waterway is calculated by means of the method developed on the basis of the modified PIANC guidelines. The main goal of the research is to prove and demonstrate that the designed software would directly increase the navigation safety level of the Świnoujście–Szczecin fairway and indicate the optimal positioning methods in various navigation circumstances.

**Keywords:** safety of navigation; safety management system; fairway; navigation channel; marine traffic engineering

## 1. Introduction

The aim of the work described in the paper was to build an application of the integrated navigation safety management system (INSMS) for coastal waters and harbour approaches in order to easily estimate the risk level of a selected part of the waterway in predefined hydrometeorological and navigation conditions. Such analyses require complex calculations, simulations, and expert knowledge [1]. They are usually carried out during a fairway's definition and deployment phases but rarely while operating ships in the fairway on a daily basis [2]. The premise of the developed computer application was that it should support the daily work and decision processes of the persons responsible for the vessel traffic management in the approach and harbour waters. On the other hand, such a system could likewise be a very useful tool for designing new waterways and modifying existing ones. Due to different manoeuvring limits in various types of water areas, the INSMS for restricted water areas (approaches to ports) requires a systematic approach to the safety assessment [3]. Such an approach covers the use of models of navigational risk assessments appropriate and adequate to the specific navigation conditions. The developed INSMS application codenamed "NEPTUN" includes the following modules representing the various aspects analysed in terms of the safety level assessment (Figure 1):

(1)  marine accident statistics module;
(2)  questionnaire survey module;
(3)  waterway calculation module for a given part of fairway;
(4)  risk analysis module;
(5)  recommendation module for the route being implemented based on the incident analysis;
(6)  legal regulation analysis module.

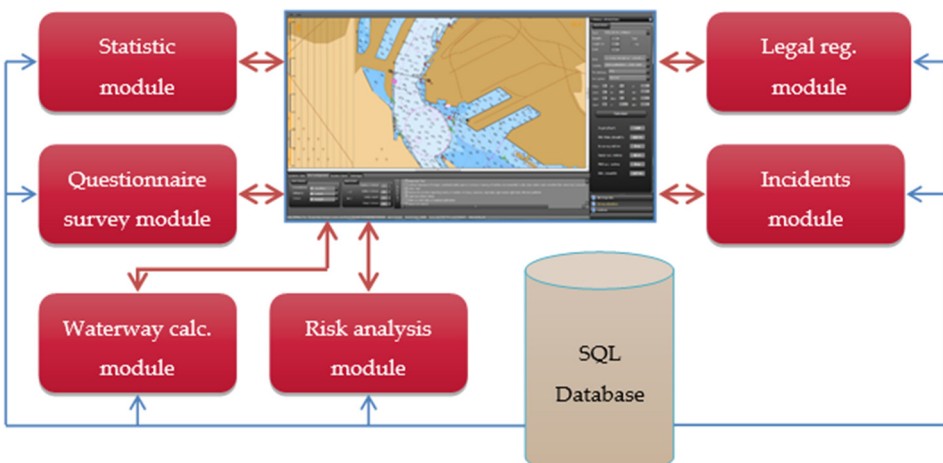

**Figure 1.** INSMS "NEPTUN" application flowchart.

All of the above modules use data stored in the database of the SQL server. Such a solution is dictated by their large number and the fact that some of the computations, due to their complexity and time consumption, are performed in the data server as well. The results of these calculations are, in turn, sent to the client application installed on the end user's computer. The application is dedicated mainly to port authorities as a tool assisting in the navigation safety assessment of a fixed segment of a waterway. As well as related proposals of navigation decision systems assisting in anticollision manoeuvres in open waters [4], "NEPTUN" can be applied during the post-factum analysis.

The INSMS "NEPTUN" is used for assessment of the navigation or manoeuvring risk as an advisory system for traffic control services, but it is also possible to use its individual modules to carry out specific research. The module for determining the width of the safe manoeuvring area was used in this study, presented in Sections 3–5. This investigation concerned the possibility of allowing a bulk carrier of parameters exceeding the ones set in the harbour regulations to enter the port of Szczecin in Poland. As the result of this study, a set of waterway parameters was obtained as the basis for the consecutive process leading to changes of the aids in navigation or even reconstruction of some parts of the fairway.

## 2. Related Works

There has been significant progress within the last decades in the development of safety management and decision support systems [5,6]. In the 1980s, deterministic methods of designing the parameters of sea waterways were still dominant [7]; in the 1990s and the first decade of the 21st century, there was a development of probabilistic methods with the use of simulation tests [8]. The vessel technical error (VTE) was introduced by IMO in 2001 [9] and used in research [10]. The systematic approach to the waterway design has begun to be introduced nowadays [3] and was applied during the INSMS "NEPTUN" application development presented in this paper.

In terms of marine traffic engineering (MTE), the sea waterway is a restricted area of water adapted and used for the ship traffic of various vessel types. MTE deals with the qualitative and quantitative analyses of this traffic in areas limited in terms of navigation safety. The MTE system is in a safe state if it allows the accident-free passage of the ship in accordance with her route plan while maintaining the required traffic parameters, i.e., the safety level of the MTE system is related to the probability of accident-free passage of the ship or to the accident risk (probability of a ship accident in the domains of time and space and its consequences). A Quantitative Risk Assessment (QRA), as a formal and systematic risk analysis approach to estimating the likelihood and consequences of hazardous events and expressing the results quantitatively as a risk, was introduced to MTE in the 1990s [11]. Nowadays, it has become the marine engineering standard introduced by DNV [1] and others [12,13]. Its full introduction into the fairway operation on a daily basis has not been

implemented yet [2]. The numerical basics of the final QRA in the application developed by the authors have been adopted from the Polish method [3] used for calculating the safe widths of fairways, which evolved from a number of empirical deterministic methods, like: PIANC [14], Spanish [15], Japanese [2], USACE [16], and Canadian [17], after the supplementation of probabilistic/statistical terms and the validation of used models by simulation studies in a full mission bridge simulator.

It is worth mentioning the wider aspect of a waterway system's safety. The authors that presented the research focused on the waterway parameters, type of the vessel, and their size i.e., length, breadth, and draft. The propulsion system was not taken into account, but it is of great importance in the context of the manoeuvrability of the vessel—in particular, modifications affecting the vessel's behaviour in shallow waters [18–20]. The same aspect concerns the shape of the hull [21]. Moreover, taking into account the technological development, one cannot forget about the research on remote piloting while approaching the port via an intelligent fairway [22], as well as the current impact of the hydrometeorological conditions on the ship's behaviour on the fairway [23].

## 3. Scope of Research

It is assumed before starting the design of a new or modernisation of an existing waterway that both the maximum width of the fairway and its depth are determined, taking into account, apart from the characteristic vessel parameters like overall length, breadth, maximum height above the loading line, and maximum draft, the following factors:

(1)   maximum speeds of the vessels, taking into account their types and dimensions;
(2)   maximum speeds and directions of the prevailing winds, especially crosswinds;
(3)   maximum speeds and directions of the prevailing lateral and longitudinal currents;
(4)   maximum parameters of the wind wave mainly affecting the vessels sailing on sea waterways in the open sea or inland waters.

Regardless of the above, it is advisable when determining the dimensions of a fairway to take into account the manoeuvrability of the ship, the level of risk from the cargo, and the intensity of ship's passages through the considered waterway or its element. It is extremely important to make a forecast of the changes to the parameters of the fairway during its use, resulting from the decrease in the depth and even the width, as a result of:

(1)   sediment depositing on the slopes and the bottom of the fairway, carried or dragged by the water current; this deposition intensifies during storm surges, backwater, and very low water levels;
(2)   gathering of sediments at the bottom coming from adjacent waters reservoirs, usually very shallow;
(3)   sliding of fairway slopes to the bottom of the fairway as a result of the interactions of the currents and waves caused by vessels.

The probabilistic part covers the position, navigation, and timing (PNT) data accuracy and resultant VTE, which are quite substantial, while monitoring ships remotely only by electronic means.

This research covered the Świnoujście–Szczecin fairway channel, its navigation infrastructure, and the limitations to the bulk carrier parameters entering the port of Szczecin. The need to conduct such a research originated from the plans of the port infrastructure expansion and deepening of the waterway to a depth of 12.5 m. This study was conducted in 25 distinctive sections of Świnoujście–Szczecin (Table 1, Figures 2 and 3).

**Table 1.** List of the tested sections at the Świnoujście–Szczecin fairway.

| Section No. | Name | Section Length [m] | Description |
|---|---|---|---|
| 1 | Świnoujście Entrance | 3200 | from 0.5 km north approaching the fairway to Świnoujście (A buoy) to 2.7 km Świnoujście–Szczecin fairway |
| 2 | Kosa Turn | 2700 | from 2.7 km to 5.4 km (Light on N Cape of Mielin Island) (entrance in Mielin S leading lights) |
| 3 | Mielieński Canal | 875 | from 5.4 km to 6.275 km (Lignt No1 on Mielieński Canal) (exit from Mielin S leading lights) |
| 4 | Mielin Turn | 1275 | from 6.275 km to 7.55 km (Light No4 on Mieleński Canal) (entrance in Mielin N-Paprotno leading lights) |
| 5 | Karsibórz Ferry | 2600 | from 7.55 km to 10.2 km (Light No1 Paprotno) (exit form Mielin N -Paprotno leading lights) |
| 6 | Paprotno Turn | 900 | from 10.15 km to 11.1 km Light on W coast of Piastowski Canal (entrance in Karsibórz leading lights) |
| 7 | Piastowski Canal | 5600 | from 11.1 km to 16.8 km (breakwater light at exit from Piastowski Canal) |
| 8 | Zalew Szczeciński "A" part | 5100 | from 16.8 km to 21.9 km (special buoy at BT II anchorage) |
| 9 | Zalew Szczeciński "B" part | 4600 | from 21.9 km to 26.45 km (pair of buoys no. 5/no. 6) |
| 10 | Zalew Szczeciński "C" part | 3400 | from 26.45 km to BT III |
| 11 | Zalew Szczeciński "D" part | 5700 | from BT III to 35.5 km (pair of buoys no. 9/no. 10) |
| 12 | Zalew Szczeciński "E" part | 4600 | from 35.5 km to 40.1 km (pair of buoys no. 17/no. 18) |
| 13 | Zalew Szczeciński "F" part | 1250 | from 40.1 km to 41.36 km (pair of buoy no. 19/beacon no. 20) (exit from Mańków leading lights) |
| 14 | Żuławy Turn | 1040 | from 41.36 km to 42.4 km (buoy no. 23 / beacon no. 24). (entrance in Stepnica-Raduń leading lights) |
| 15 | Mańków | 3900 | from 42.4 km to 46.3 km (buoys 33 /34) (transition from Stepnica-Raduń leading lights to Krępa-Domańce leading lights) |
| 16 | Szeroki Nurt | 2700 | from 46.3 km to 49 km (exit from Krępa/Domańce leading lights) |
| 17 | Police Turn | 1500 | from 49 km to 50.5 km (dolphin no. 36) (entrance in Police/Ina leading lights) |
| 18 | Domiąża | 1500 | from 50.5 km to 51.956 km (buoy no. 39/dolphin no. 40) |
| 19 | Mnisi Turn | 1200 | from 51.956 km to 53.2 km (buoy no. 43/dolphin no. 44) (entrance in Łąki/Bykowo leading lights) |
| 20 | Skolwiński Ostrów | 1400 | from 53.23 km to 54.6 km (North dolphin at Żurawia Island) (exit form Łąki/Bykowo leading lights) |
| 21 | Wyspa Żurawia Turn | 1400 | from 54.6 km to 56.0 km (dolphin no. 48) (entrance in Ina S/Święta leading lights) |
| 22 | Stołczyn | 3600 | from 56.0 km to 59.6 km (dolphin no. 66) (exit form Ina S/Święta leading lights) |
| 23 | Radolin Turn | 4500 | from 59.6 km to 64.1 km (pair of dolphins no. 87/88) |
| 24 | Przekop Mieleński | 2700 | from 64.1 km to 66.8 km (pair of dolphins no. 99/100) |
| 25 | Parnica Turn | 700 | from 66.8 km to 67.5 km (dolphin no. 104) |

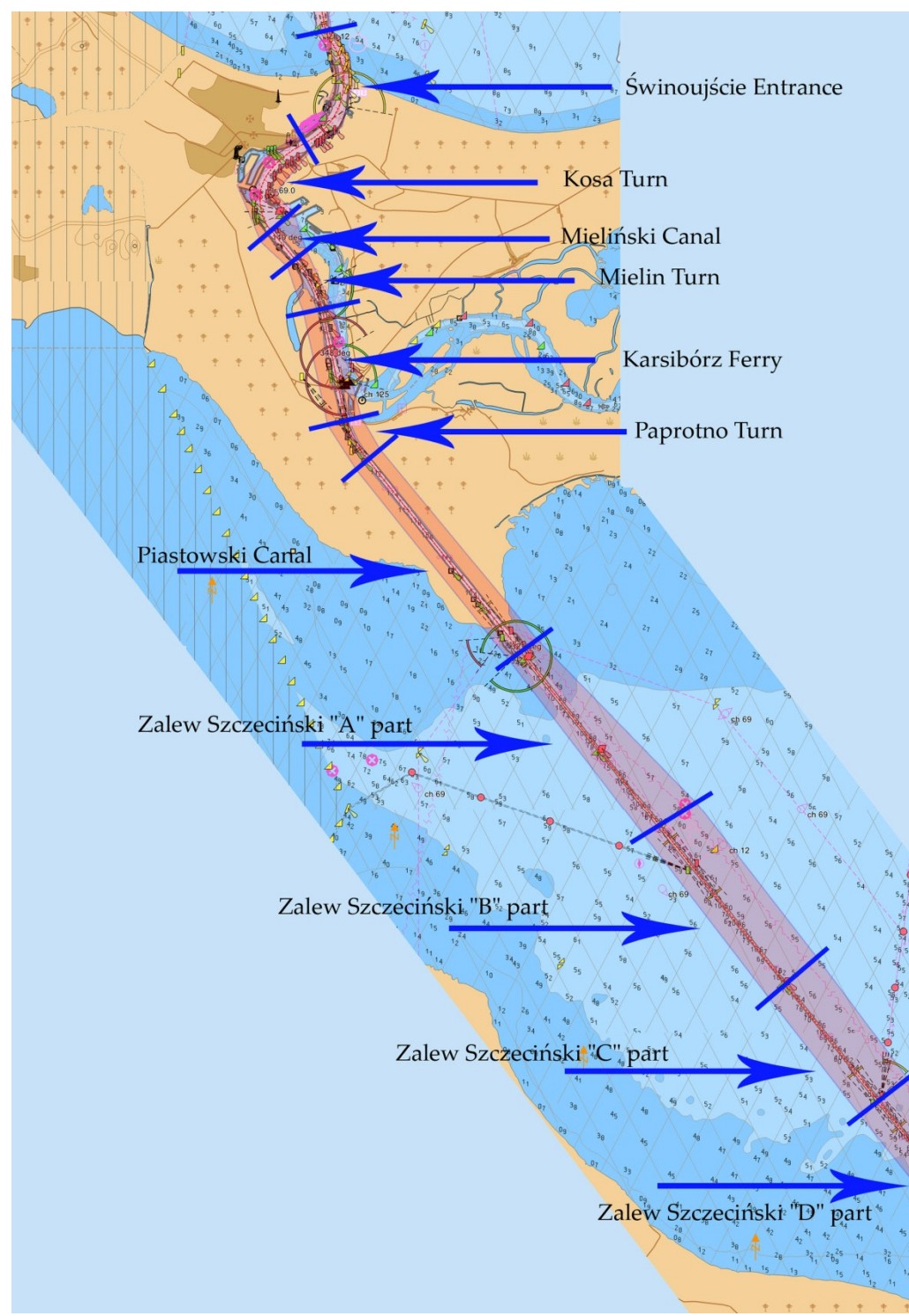

**Figure 2.** Areas of the tested sections at the Świnoujście–Szczecin fairway, northern part.

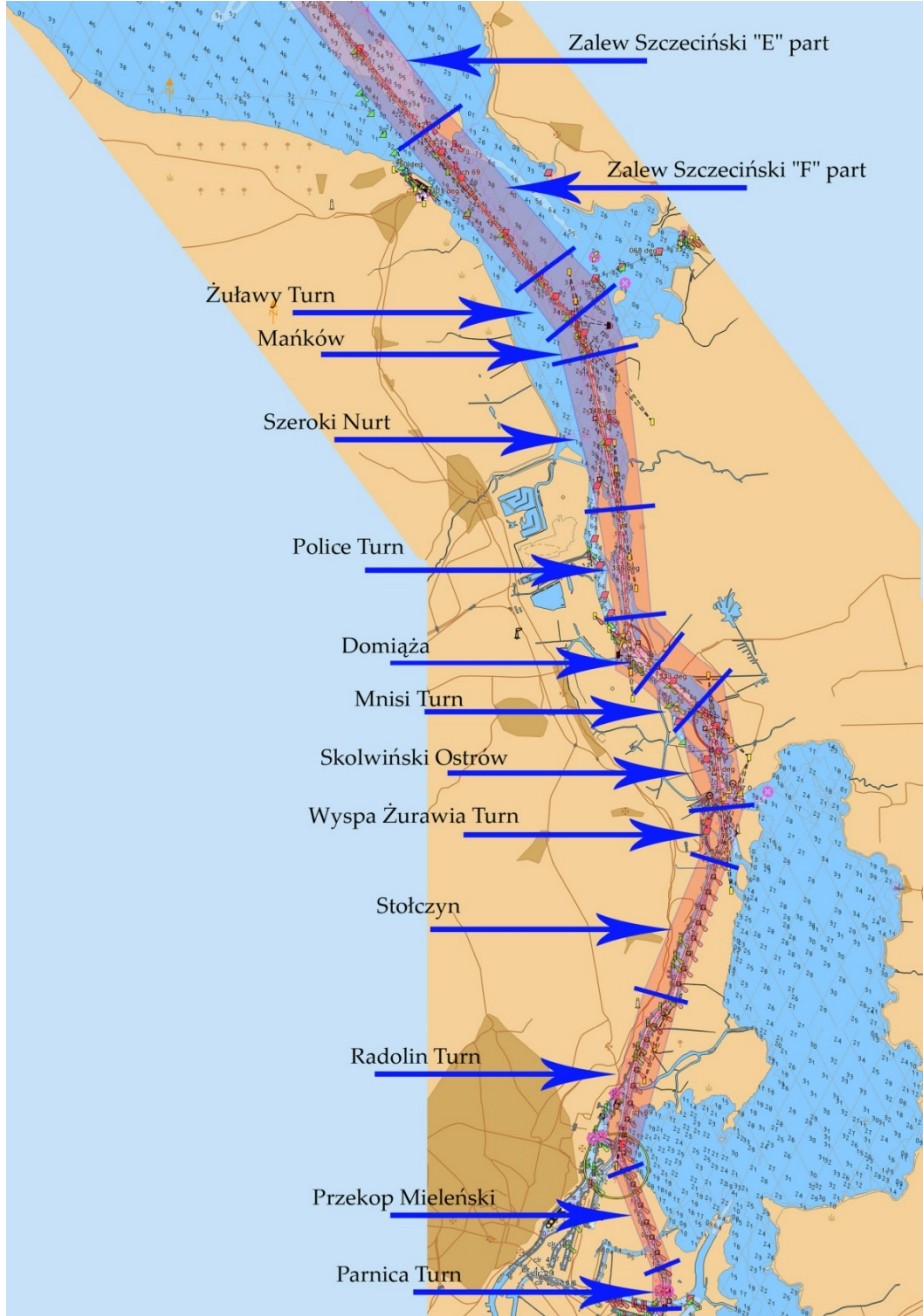

**Figure 3.** Areas of the tested sections at the Świnoujście–Szczecin fairway, southern part.

## 4. Materials and Methods

The Polish method of water channel design was developed at the Maritime University of Szczecin [3,24]. It is based on the minimising of the objective function related to the parameters of a fairway composed of one-way and two-way sections with respect to the fairway reconstruction and operational costs, taking into account the costs of delays of ships belonging to specific size groups for projected traffic intensities [25]. In this study, due to the maximum parameters of the tested vessel, two-way traffic was not taken into account. However, the specific fairway types were taken into consideration, divided into straight sections and bends [26].

Taking into consideration the one-way traffic, the safety passage is performed under the following conditions:

$$d_j(1 - \alpha) + d_r^p + d_r^l \leq D_j \tag{1}$$

where: $D_j$: width at the bottom of the *j*th point of the fairway (axis) for a safe depth [m]; $d_j(1-\alpha)$: width of the safe manoeuvring area at the given confidence level $(1-\alpha)$ for the *j*th point of the fairway (axis) (m); $d_r^p$: fairway width reserve for the bank–channel effect on the right side of the fairway (m); $d_r^l$: fairway width reserve for the bank–channel effect on the left side of the fairway (m).

The width of the safe manoeuvring area is calculated according to the formula

$$d_j(1-\alpha) = \overline{X_{jp}} - \overline{X_{jl}} + c\sigma_{jp} + c\sigma_{jl}\,[\text{m}] \tag{2}$$

where: $c$: confidence factor; $\overline{X_{jp}}; \overline{X_{jl}}$: arithmetic mean of the vessel maximum port and starboard distances from the *j*th point of the fairway (m); $\sigma_{jp}; \sigma_{jl}$: standard deviations of the vessel maximum port and starboard distances from the *j*th point of the fairway (m).

On a straight fairway, we can assume that

$$\overline{X_{jp}} = \left|\overline{X_{jl}}\right| = \overline{X_j} \tag{3}$$

and

$$\sigma_{jp} + \sigma_{jl} = \sigma_j \tag{4}$$

Taking the above into account, we have

$$d_j(1-\alpha) = 2(\overline{X_j} + c\sigma_j)\,[\text{m}] \tag{5}$$

and, finally,

$$D_j \geq 2\overline{X_j} + 2c\sigma_j + d_r^p + d_r^l \tag{6}$$

In the Polish method [3], the manoeuvring component of the manoeuvring area (fairway lane) width is defined deterministically, while the navigational component of the manoeuvring area's width is probabilistic and is determined at a certain confidence level. Taking this into account, the width of the safe manoeuvring area for a straight waterway section is determined as follows:

$$d(1-\alpha) = d_m + 2d_n(1-\alpha)\,[\text{m}] \tag{7}$$

The results of the research carried out have shown that, to determine the mean width of the manoeuvring area, some modifications of the deterministic PIANC method can be used. In this method, the width of the safe manoeuvring area for straight fairways is determined according to the following equation [13]:

$$d = d_{mp} + \sum_{i=1}^{9} d_i + d_r^p + d_r^l\,[\text{m}] \tag{8}$$

where: $d_{mp}$: basic manoeuvring width of the traffic lane (m_; $d_i$: additional corrections of the traffic lane width: $i = 1$, vessel speed; $i = 2$, prevailing transverse wind; $i = 3$, prevailing transverse current; $i = 4$, prevailing longitudinal current; $i = 5$, height and length of the significant wave; $i = 6$, navigational markings and traffic regulation system; $i = 7$, type of bottom; $i = 8$, the ratio of the depth to the draft of the vessel; $i = 9$, the risk caused by the cargo carried. $d_r^l$: lane width reserve on the left (m); $d_r^p$: lane width reserve on the right (m).

The basic width of the traffic lane is determined depending on the vessel's steerability: $d_{mp}$ = 1.3 $B$—very good steerability; $d_{mp}$ = 1.5 $B$—good steerability; $d_{mp}$ = 1.8 $B$—poor steerability.

In the Polish method [3], it was assumed that the arithmetic mean of the maximum distances of the ship's point to the right and left from the track axis can be determined deterministically with some approximations, according to the following formulas:

For the vessel manoeuvring alone:

$$2\overline{X_j} \approx d_m = d_{mp} + d_1 + d_2 + d_3 + d_4 + d_5 + d_7 + d_8 \tag{9}$$

For the vessel manoeuvring with the tugs:

$$2\overline{X_j} \approx d_m = d_{mp} + d_1 + d_2 + d_3 + d_4 + d_5 + d_7 \tag{10}$$

where $d_m$: manoeuvring component of the safe manoeuvring area (m).

Finally, it was assumed that the standard deviation of the maximum distances of the points to the right and left of the fairway axis approximates the directional error of the ship's side position to the right and left determined at the appropriate confidence level. We call it the navigational component of the safe manoeuvring area, which is

$$c\sigma_{jp} = d_n^p(1 - \alpha) \tag{11}$$

$$c\sigma_{jl} = d_n^l(1 - \alpha) \tag{12}$$

Taking into account the above assumptions, the width of the manoeuvring area on rectilinear fairways can be determined as follows (Figures 1 and 4):

$$d(1 - \alpha) = d_m + d_n^p(1 - \alpha) + d_n^l(1 - \alpha)[m] \tag{13}$$

The navigation components of the safe manoeuvring area (right and left) in rectilinear fairways are as follows:

$$d_n^p(1 - \alpha) = d_n^l(1 - \alpha) = d_n(1 - \alpha) \tag{14}$$

i.e., the width of the manoeuvring area at the confidence level $(1 - \alpha)$ is

$$d(1 - \alpha) = d_m + 2d_n(1 - \alpha)[m] \tag{15}$$

where $d(1 - \alpha)$: safe width of the manoeuvring area at the confidence level (m); $D$: width of the available shipping area (m); $d_m$: manoeuvring component of the width of the safe manoeuvring area (m); $d_n(1 - \alpha)$: the navigation component of the safe manoeuvring area at the confidence level $(1 - \alpha)$ (m); $d_n^p(1 - \alpha)$; $d_n^l(1 - \alpha)$: the navigation components (left and right) of the width of the safe manoeuvring area at the confidence level $(1 - \alpha)$(m).

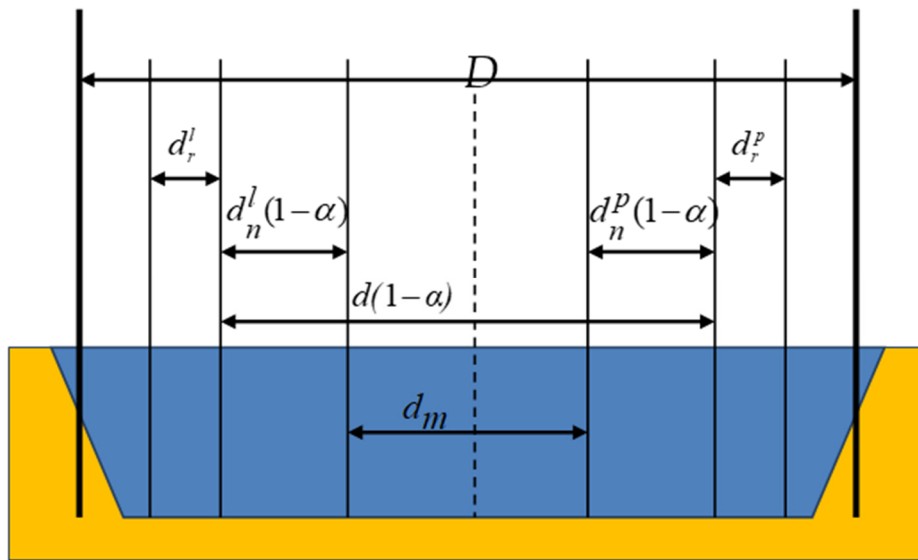

**Figure 4.** Safe width of the manoeuvring area at a certain confidence level (one-way fairway) [3].

The safe widths of the manoeuvring area in the fairways for each of the selected navigation systems must meet the following conditions:

$$D \geq d(1-\alpha) + d_r^p + d_r^l = d_m + 2d_n^k(1-\alpha) + d_r^p + d_r^l \tag{16}$$

where $D$: the width of the fairway at the bottom for a safe depth (available fairway depth); $d_n^k(1-\alpha)$: the navigation component of the safe manoeuvring area for the $k$th position system at the appropriate confidence level for that system $(1-\alpha)$.

If the right and left manoeuvring area width reserves are equal (equal fairway slopes):

$$d_r^p = d_r^l = d_r \tag{17}$$

Therefore, the conditions for safe navigation can be written as follows:

$$D \geq d(1-\alpha) + 2d_r = d_m + 2d_n^k(1-\alpha) + 2d_r \tag{18}$$

where $d_r; d_r^p; d_r^l$: reserve of the manoeuvring area width (m).

The fairway width for the bank clearance, taking into account the edge and channel effect, is determined depending on the speed of the ship and the type of the slope according to the PIANC methodology [13].

The navigation component of the manoeuvring area in a straight fairway for a given positioning system is the directional error of the ship's side position determined at a given confidence level. This is the directional error perpendicular to the fairway axis, which is [14]

$$d_n(1-\alpha) = p_{yB}(1-\alpha) = \pm\sqrt{p_y(1-\alpha)^2 + \left(\frac{m_{KR}^B(1-\alpha)L_D}{57.3°}\right)^2} \; [\text{m}] \tag{19}$$

where

$p_{yB}(1-\alpha)$—directional error of the ship's side at the confidence level $(1-\alpha)$[m];

$p_y(1-\alpha)$—directional error in determining the vessel's position (observer's position) at the confidence level $(1-\alpha)$[m];

$m_{KR}^B(1-\alpha)$—error in determining and maintaining the ship's course on a straight section of the fairway, confidence level $(1-\alpha)$ (°);

$L_D$—the distance between the bow and the bridge of the ship [m].

For a "maximum vessel" with a superstructure in the aft part, it was assumed: $m_{KR}^B(0.95) = 1°$: PNS (Pilot Navigation System); $m_{KR}^B(0.95) = \pm2°$: the rest of the navigational positioning systems; $L_D = 0.75L_c$.

When determining the position using floating aids to navigation like buoys, the error of the position of the buoy in relation to the position of its anchor should be taken into account:

$$p_{yp}(0.95) = kh_p \tag{20}$$

where

- directional error of the buoy deflection perpendicular to the fairway axis at the confidence level 0.95;
- buoy anchorage depth (m);
- coefficient that depends on the wave height in the given water area.

At the bend of the fairway for one-way traffic, the width of the safe manoeuvring area at the confidence level is determined according to the relationship

$$d_z(1-\alpha) = d_{mz} + 2d_n(1-\alpha) \tag{21}$$

where the manoeuvring component of the safe manoeuvring area width is, respectively,

$$d_{mz} = d_m + \Delta d \tag{22}$$

where

$d_z(1-\alpha)$—safe width of the manoeuvring area for a bend at the confidence level $(1-\alpha)$ [m];
$d_{mz}$—manoeuvring component of the width of the safe manoeuvring area in a bend [m];
$d_m$—manoeuvring component of the width of the safe manoeuvring area on a straight section [m];
$\Delta d$—widening of the ship's lane in the bend (m).

The ship's lane widening in the bend is determined using the modified Canadian method for determining the width of fairways. The widening of the ship's lane $\Delta$d in the bend is taken into account in cases where the planned radius of the bend R is less than $10L_c$ ($L_c$—total length of the ship) according to the Dave Taylor dependence adopted in the Canadian method [13].

$$\Delta d = \frac{3.4451\Delta\psi V^2 L_c^2 F}{Rk_z s} \text{ [m]} \tag{23}$$

where

$\Delta\psi$—turning angle (course change) on a bend (°);
$V$—vessel speed (m/s);
$F$—a coefficient of 1 for one-way traffic, 2 for two-way traffic;
$R$—radius of the arc in the bend (m);
$s$—the minimum required visibility from the ship's bridge $\geq$2446 m (m);
$k_z$—the ship's steering coefficient (1—poor, 2—good, and 3—very good).

The required bend radius $R$ is determined on the basis of empirically determined dependencies of the ratio of the bend radius to the ship length on the basis of the interpolated spline curves. The recommended radius of the bend curved with the help of tugs was set at a minimum of 3L [14]:

$$p_n(1-\alpha) = p_{yD}(1-\alpha) = \sqrt{p_y(1-\alpha)^2 + \left(\frac{m_{KR}^D(1-\alpha)L_p}{57.3°}\right)^2} \text{ [m]} \tag{24}$$

where

$p_{yD}(1-\alpha)$—directional error of the ship's bow position at the confidence level $(1-\alpha)$ [m];
$p_y(1-\alpha)$—directional error in determining the vessel's position (observer's position) at the confidence level $(1-\alpha)$ [m];
$m_{KR}^D(1-\alpha)$—error in determining the ship's course in a bend, confidence level $(1-\alpha)$ [°].

For the "maximum vessel" (superstructure at the aft), it was assumed: $m_{KR}^D(0.95) = 2°$: PNS (Pilot Navigation System); $m_{KR}^D(0.95) = 4°$: the rest of the navigational positioning systems; $L_D = 0.75L_c$.

### 4.1. Vessel and Area Parameters

The following research conditions were adopted:

(1)  Maximum vessel: Bulk carrier: $L_c$ = 230 m, $B$ = 36.0 m, $T$ = 11.0 m
(2)  Under keel clearance: $\Delta$ = 1.5 m
(3)  Maximum vessel speed limit: V $\leq$ 8 knots
(4)  Tug assistance: none
(5)  Permissible hydrometeorological conditions: Time of day: no limits

Wind: speed up to 10 m/s, direction—no limits
Current: speed up to 1 knot, direction—along the fairway
Wave: none
Visibility: above 2 NM

(1)  Navigation system used: Good visibility:

(a)  day

(b)      PNS

(c)      Terrestrial

(2)    Good visibility:

       (a)      night

       (b)      PNS

       (c)      Terrestrial

(3)    Restricted visibility:

       (a)      Z < 2 Mm

       (b)      PNS

       (c)      Radar

Taking all the above data into consideration, the sample calculations checking the correct operation of the algorithm are presented further. Firstly, the positioning method, time of day, and the visibility are set:

(1)    positioning method: terrestrial based on lateral navigation marks;

(2)    day;

(3)    good visibility Z > 2 NM.

Next, the other factors according to Equation (8):

(1)    $d_1$—vessel speed = 8 kn;

(2)    $d_2$—prevailing transverse wind = 0 kn;

(3)    $d_3$—prevailing transverse current = 0 kn;

(4)    $d_4$—prevailing longitudinal current = 1 kn;

(5)    $d_5$—height and length of the significant wave, $h_W$ = 1.0 m, $\lambda_W$ = 50.0 m;

(6)    $d_7$—type of bottom = soft;

(7)    $d_8$—the ratio of the depth to the draft of the vessel h/T = 1.136;

Finally, the data regarding the water area:

(1)    r (m)—distance between beacons;

(2)    x (m)—distance to the closest beacon;

(3)    h (m)—depth of the water area;

(4)    type of the coast: dredged.

According to Equation (18), the width of the fairway at the bottom for a safe depth is as follows (for the straight parts of the fairway):

$$D = d_m + 2d_n^k(1 - \alpha) + 2d_r \tag{25}$$

$$d_m = d_{mp} + d_1 + d_2 + d_3 + d_4 + d_5 + d_7 + d_8 \tag{26}$$

$$d(1 - \alpha) = \sqrt{\left[1.16 \times 10^{-3}\left(1 + \frac{x}{r}\right)\right]^2 + \left(\frac{1.5L_c}{57.3}\right)^2} \; [\text{m}] \tag{27}$$

$d_{mp}$ = 1.5 B (good steerability) = 54.00 m;

$d_m$, according to Table 2, is even $d_m$ = 54.00 + 0 + 0 + 0 + 0 + 0 + 0 + 18.00 = 72.00 m:

$d_r$, according to Table 3, is as $d_r$ = 0.5 × B = 0.5 × 36.00 = 18.00 m;

$$d(1 - \alpha) = \sqrt{\left[1.16 \times 10^{-3}\left(1 + \frac{1600}{3200}\right)\right]^2 + \left(\frac{1.5 \times 230}{57.3}\right)^2} = 6.63 \text{m} \tag{28}$$

where r is taken from Table 4, and × is taken as r/2 as the worst case. The width of the fairway at the bottom for a safe depth is

D = 72.00 + 2 × 6.63 + 2 × 18.00 = 121.26 m

**Table 2.** Additional widths of the traffic lanes for the straight channel sections derived from [24].

| | Type of Correction | Vessel Speed (kn) | Outer Channel (Open Water) | Inner Channel (Protected Water) |
|---|---|---|---|---|
| | Vessel speed (kn, through the water): | | | |
| $d_1$ | $V > 12$ | | 0.1 $B$ | 0.1 $B$ |
| | $V = 8$–12 | | 0.0 | 0.0 |
| | $V = 5$–8 | | 0.0 | 0.0 |
| | Prevailing cross wind (kn): | | | |
| $d_2$ | light $\leq$ 15 kn ($\leq$Beaufort 4) | $V > 12$ kn | 0.1 $B$ | 0.1 $B$ |
| | | $V = 8$–12 kn | 0.2 $B$ | 0.2 $B$ |
| | | $V = 5$–8 kn | 0.3 $B$ | 0.3 $B$ |
| | moderate 15–33 kn (Beaufort 4–7) | $V > 12$ kn | 0.3 $B$ | 0.3 $B$ |
| | | $V = 8$–12 kn | 0.4 $B$ | 0.4 $B$ |
| | | $V = 5$–8 kn | 0.6 $B$ | 0.6 $B$ |
| | strong 33–48 kn (Beaufort 7–9) | $V > 12$ kn | 0.5 $B$ | 0.5 $B$ |
| | | $V = 8$–12 kn | 0.7 $B$ | 0.7 $B$ |
| | | $V = 5$–8 kn | 1.1 $B$ | 1.1 $B$ |
| | Prevailing cross current (kn): | | | |
| $d_3$ | negligible < 0.2 | all | 0.0 | 0.0 |
| | low 0.2–0.5 kn | $V > 12$ kn | 0.2 $B$ | 0.1 $B$ |
| | | $V = 8$–12 kn | 0.25 $B$ | 0.2 $B$ |
| | | $V = 5$–8 kn | 0.3 $B$ | 0.3 $B$ |
| | moderate 0.5–1.5 kn | $V > 12$ kn | 0.5 $B$ | 0.4 $B$ |
| | | $V = 8$–12 kn | 0.7 $B$ | 0.6 $B$ |
| | | $V = 5$–8 kn | 1.0 $B$ | 0.8 $B$ |
| | strong 1.5–2.0 kn | $V > 12$ kn | 1.0 $B$ | – |
| | | $V = 8$–12 kn | 1.2 $B$ | – |
| | | $V = 5$–8 kn | 1.6 $B$ | – |
| | Prevailing longitudinal current (kn): | | | |
| $d_4$ | low $\leq$ 1.5 kn | all | 0.0 | 0.0 |
| | moderate 1.5–3.0 kn | $V > 12$ kn | 0.0 | – |
| | | $V = 8$–12 kn | 0.1 $B$ | 0.1 $B$ |
| | | $V = 5$–8 kn | 0.2 $B$ | 0.2 $B$ |
| | strong > 3.0 kn | $V > 12$ kn | 0.1 $B$ | – |
| | | $V = 8$–12 kn | 0.2 $B$ | 0.2 $B$ |
| | | $V = 5$–8 kn | 0.4 $B$ | 0.4 $B$ |
| | Height ($h_F$) and length ($\lambda_F$) of wave (m): | | | |
| $d_5$ | $h_F \leq 1$ m; $\lambda_F \leq L$ | all | 0.0 | 0.0 |
| | 3 m > $h_F$ > 1 m; $\lambda_F = L$ | $V > 12$ kn | 2.0 $B$ | |
| | | $V = 8$–12 kn | 1.0 $B$ | |
| | | $V = 5$–8 kn | 0.5 $B$ | |
| | $h_F$ > 3 m; $\lambda_F$ > $L$ | $V > 12$ kn | 3.0 $B$ | |
| | | $V = 8$–12 kn | 2.2 $B$ | |
| | | $V = 5$–8 kn | 1.5 $B$ | |
| | Bottom surface: | | | |
| $d_7$ | $\frac{h}{T} \geq 1.5$ | | | |
| | $\frac{h}{T} < 1.5$ | | | |
| | soft (mud. clay) | | 0.0 | 0.0 |
| | medium (sand. gravel) | | 0.1 $B$ | 0.2 $B$ |
| | rough (stones. rocks) | | 0.2 $B$ | 0.4 $B$ |
| | Depth to draft ratio: | | | |
| $d_8$ | $\frac{h}{T} \geq 1.5$ | | 0.0 | 0.0 |
| | $\frac{h}{T} = 1.25 \div 1.5$ | | 0.1 $B$ | 0.2 $B$ |
| | $\frac{h}{T} < 1.25$ | | 0.2 $B$ | 0.4 $B$ |

**Table 3.** The fairway width for the bank clearance [24].

| Type of the Slope | Vessel Speed (kn) | Outer Channel ($d_r$) | Inner Channel ($d_r$) |
|---|---|---|---|
| Gentle underwater channel slope (1:10 or less steep) | $V > 12$ kn | 0.2 $B$ | 0.2 $B$ |
| | $V = 8$–$12$ kn | 0.1 $B$ | 0.1 $B$ |
| | $V = 5$–$8$ kn | 0.0 $B$ | 0.0 $B$ |
| Sloping channel edges and shoals | $V > 12$ kn | 0.7 $B$ | 0.7 $B$ |
| | $V = 8$–$12$ kn | 0.5 $B$ | 0.5 $B$ |
| | $V = 5$–$8$ kn | 0.3 $B$ | 0.3 $B$ |
| Steep and hard embankments, structures | $V > 12$ kn | 1.3 $B$ | 1.3 $B$ |
| | $V = 8$–$12$ kn | 1.0 $B$ | 1.0 $B$ |
| | $V = 5$–$8$ kn | 0.5 $B$ | 0.5 $B$ |

**Table 4.** Straight parts of the fairway, their lengths, and the calculated safe widths of the fairway for the given vessel and conditions.

| Straight Part of Fairway | r (m) | D (m) |
|---|---|---|
| Świnoujście Entrance | 3200.0 | 121.26 |
| Mieliński Canal | 875.0 | 120.14 |
| Karsibórz Ferry | 2650.0 | 120.89 |
| Piastowski Canal | 5700.0 | 123.60 |
| Zalew Szczeciński "A" part | 5100.0 | 122.96 |
| Zalew Szczeciński "B" part | 4550.0 | 122.41 |
| Zalew Szczeciński "C" part | 3400.0 | 121.42 |
| Zalew Szczeciński "D" part | 5650.0 | 123.55 |
| Zalew Szczeciński "E" part | 4600.0 | 122.46 |
| Zalew Szczeciński "F" part | 1260.0 | 120.24 |
| Mańków | 3900.0 | 121.82 |
| Szeroki Nurt | 2700.0 | 120.93 |
| Domiąża | 1456.0 | 120.31 |
| Skolwiński Ostrów | 1400.0 | 120.29 |
| Stołczyn | 3600.0 | 121.57 |
| Przekop Mieleński | 2700.0 | 120.93 |

The results for all straight parts of the fairway based on the above calculations are presented in Table 4.

### 4.2. Implementation of the Algorithm of the Deterministic-Probabilistic Method in NEPTUN Application

The INSMS application "NEPTUN" is mainly used for assessing the navigation risk, but in this study, the only "Waterway calculation module" was used for calculating the recommended fairway width for the particular vessel. This module work was based on the Polish method of navigation channels parameters assessment described above. For the definition of the input parameters, a user-friendly interface was designed (Figure 5).

The block diagram of the algorithm for determining the safe fairway width is presented in Figure 6.

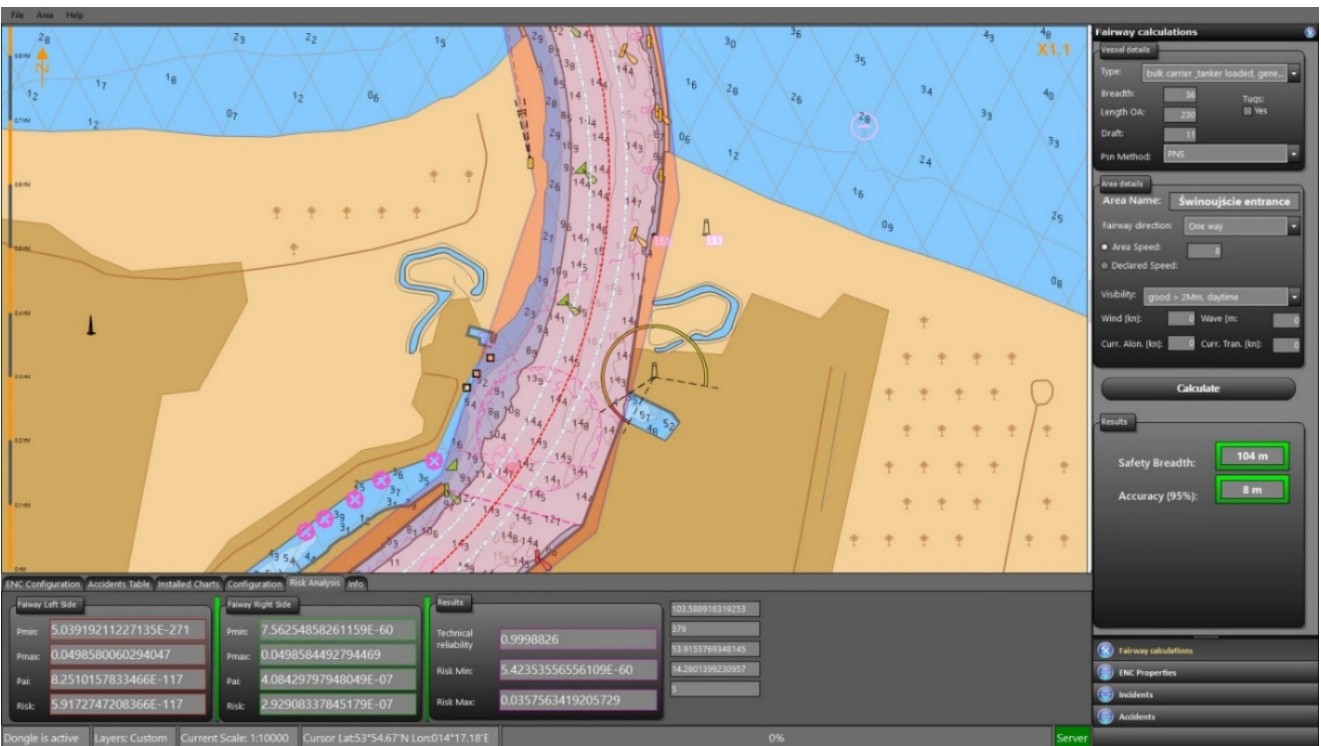

**Figure 5.** "NEPTUN" application—user interface with input field, electronic chart, and results.

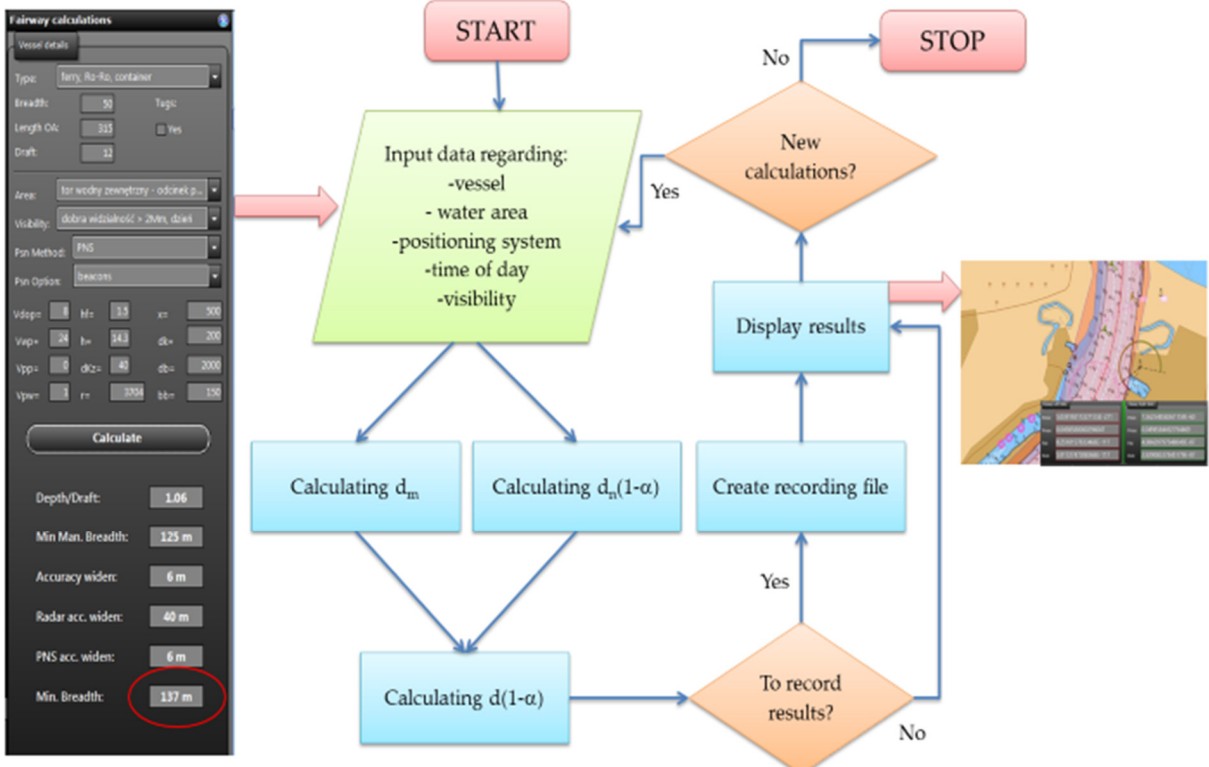

**Figure 6.** Block diagram of the algorithm for determining the safe fairway width d $(1 - \alpha)$ using the deterministic–probabilistic method.

The user interface allows the input of all the data necessary for the fairway width calculation, such as:

(1) type of vessel and its breadth, length, and draft;
(2) section of the waterway (choosing directly from the digital chart by clicking the left mouse button);
(3) visibility;
(4) positioning system and its options;
(5) allowed vessel speed;
(6) wind and current parameters.

As a result of the calculations, we get:

(1) underwater clearance;
(2) minimum width of the fairway section for the given input parameters;
(3) fairway width correction for accuracy;
(4) fairway width correction for the used positioning systems (PNS, radar, terrestrial);
(5) corrected width of the fairway section for the given input parameters at the confidence level set or confidence level achieved via the risk assessment.

### 4.3. Research Results of Determining the Width of the Fairway by INSMS "NEPTUN" Application

The safe fairway widths were determined at a confidence level of 0.95 as the maximum values of the safe manoeuvring areas for one type of "maximum vessel" two times per day (day and night) and for two positioning navigation systems provided for a given time of day.

In Table 5 there is a list of the fairway widths of the safe manoeuvring areas for the "maximum vessel" at the seabed with a depth of 12.5 m for the current navigational markings at the confidence level $(1 - \alpha) = 0.95$. Figures 7–9 present the calculated widths of the fairway, so, in Figure 7, the fairway widths are estimated for restricted visibility (Vis < 2 NM) for two positioning systems, PNS and radar. As it is clearly visible in such a situation, the PNS system has an advantage over the positions taken by radar. In Figure 8, there are conditions with good visibility and, also, two positioning systems, but this time, instead of radar, the terrestrial method is used. In this case, using the PNS system is more favourable on the straight parts of the fairway, but on the turns, during perfect visibility, the terrestrial method gives better results. In Figure 9, night-time is taken into account, and the same positioning methods (PNS, terrestrial) as during good visibility and daytime were under consideration. As one could expect, using PNS gives the lowest fairway widths for a given vessel on the straight parts of the fairway. The terrestrial method again was better for the turns, but compared to the same method during the daytime, the widths are slightly bigger. Those calculations and results were to answer the question of whether it is possible to enter the port of Szczecin under the assumed conditions using various positioning systems and under which conditions the required widening of the fairway will be the smallest.

**Table 5.** List of the fairway widths in m of the safe manoeuvring areas for the "maximum vessel" at the seabed with a depth of 12.5 m for the current navigational markings, $(1 - \alpha) = 0.95$.

| Fairway Section | Beginning of Section (km) | PNS | Restricted Visibility (Radar—Daytime) | Good Visibility (Terrestrial—Daytime) | Good Visibility (Terrestrial—Night) | Fairway Widths |
|---|---|---|---|---|---|---|
| Świnoujście Entrance | −0.5 | 120.00 | 158.03 | 121.26 | 121.91 | 150 |
| Kosa Turn | 2.7 | 146.00 | 159.62 | 133.54 | 133.61 | 150 |
| Mieliński Canal | 5.4 | 120.00 | 145.08 | 120.14 | 120.19 | 150 |
| Mielin Turn | 6.275 | 146.00 | 152.48 | 132.82 | 132.86 | 150 |
| Karsibórz Ferry | 7.55 | 120.00 | 154.94 | 120.89 | 121.35 | 150 |
| Paprotno Turn | 10.2 | 146.00 | 150.67 | 132.53 | 132.55 | 150 |
| Piastowski Canal | 11.1 | 120.00 | 172.20 | 123.60 | 125.28 | 110 |
| Zalew Szczeciński "A" part | 16.8 | 120.00 | 168.78 | 122.96 | 124.37 | 110 |
| Zalew Szczeciński "B" part | 21.9 | 120.00 | 165.66 | 122.41 | 123.59 | 110 |
| Zalew Szczeciński "C" part | 26.45 | 120.00 | 159.16 | 121.42 | 122.13 | 110 |
| Zalew Szczeciński "D" part | 29.85 | 120.00 | 171.91 | 123.55 | 125.21 | 110 |
| Zalew Szczeciński "E" part | 35.5 | 120.00 | 165.95 | 122.46 | 123.66 | 110 |
| Zalew Szczeciński "F" part | 40.1 | 120.00 | 147.20 | 120.24 | 120.35 | 110 |
| Żuławy Turn | 41.36 | 146.00 | 151.34 | 132.82 | 132.86 | 150 |
| Mańków | 42.4 | 120.00 | 161.98 | 121.82 | 122.73 | 150 |
| Szeroki Nurt | 46.3 | 120.00 | 155.22 | 120.93 | 121.40 | 150 |
| Police Turn | 49 | 146.00 | 153.58 | 132.82 | 132.86 | 150 |
| Domiąża | 50.5 | 120.00 | 148.29 | 120.31 | 120.45 | 110 |
| Mnisi Turn | 51.956 | 146.00 | 152.33 | 132.82 | 132.86 | 150 |
| Skolwiński Ostrów | 53.2 | 120.00 | 147.98 | 120.29 | 120.42 | 110 |
| Wyspa Żurawia Turn | 54.6 | 146.00 | 153.09 | 132.82 | 132.86 | 150 |
| Stołczyn | 56 | 120.00 | 160.29 | 121.57 | 122.36 | 110 |
| Radolin Turn | 59.6 | 146.00 | 169.05 | 134.24 | 134.34 | 150 |
| Przekop Mieleński | 64.1 | 120.00 | 155.22 | 120.93 | 121.40 | 110 |
| Parnica Turn | 66.8 | 146.00 | 157.08 | 133.54 | 133.61 | 150 |

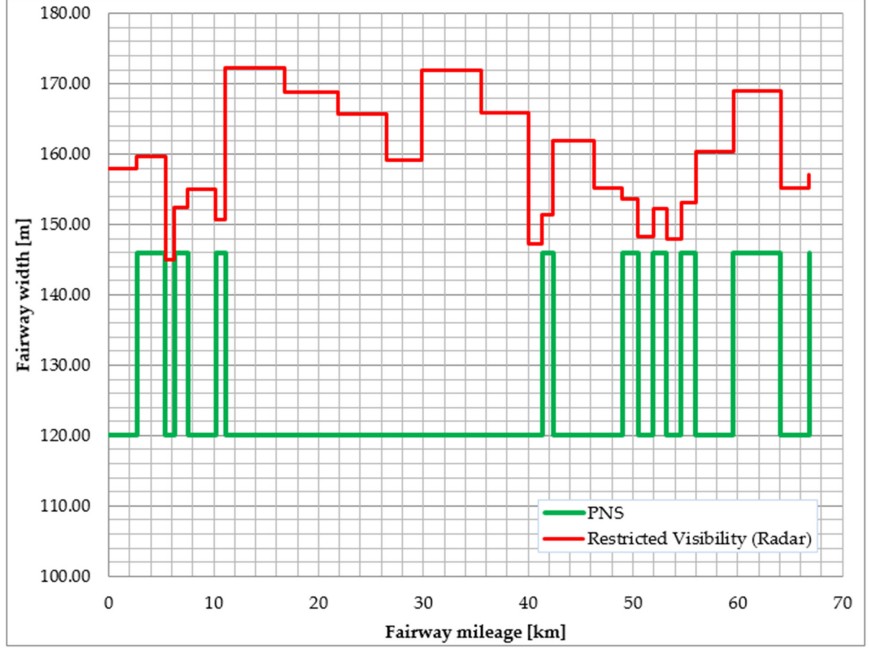

**Figure 7.** Fairway widths of the safe manoeuvring areas for a bulk carrier going from Świnoujście to Szczecin with restricted visibility and two positioning systems (PNS, radar).

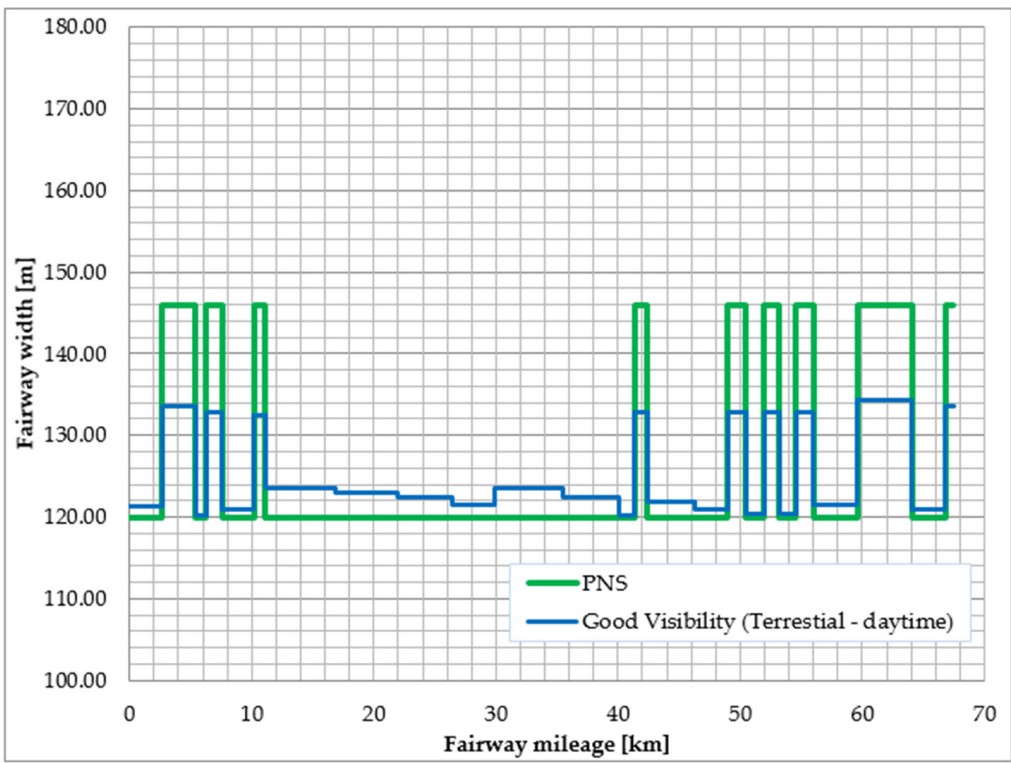

**Figure 8.** Fairway widths of the safe manoeuvring areas for a bulk carrier going from Świnoujście to Szczecin with good visibility and two positioning systems (PNS, terrestrial).

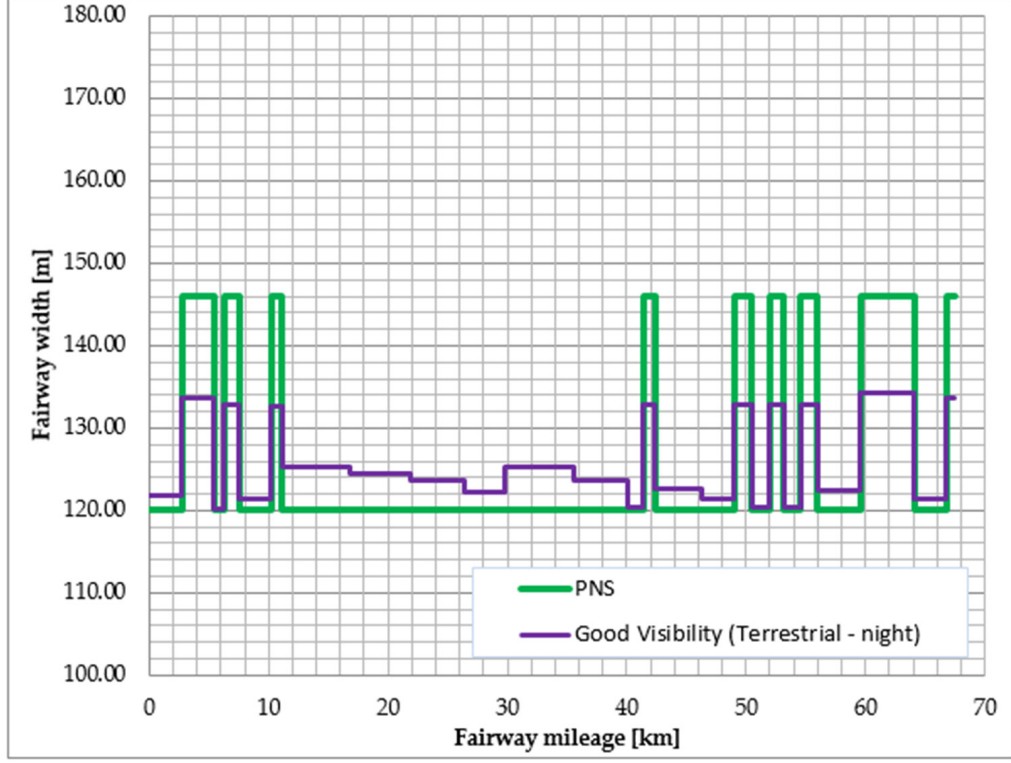

**Figure 9.** Fairway widths of the safe manoeuvring areas for a bulk carrier going from Świnoujście to Szczecin with good visibility at night with two positioning systems (PNS, terrestrial).

### 5. Discussion

The width of the safe manoeuvring area is set at a certain level confidence, which depends on the types of manoeuvres performed, their intensities, and the adopted levels of risk [16]. In practice, the decisive element is the lane width reserve, depending on the

(1) accepted confidence level (acceptable risk);
(2) traffic intensity;
(3) water parameters;
(4) parameters of the manoeuvre being performed.

The simulation and analytical research were carried out for a dredged fairway up to 12.5 m in the section from 5.0 km to 66.0 km; this made it possible to determine a safe fairway width for ships with the maximum parameters [20]:

(1) bulk carriers and container ships (L = 220 m, B = 32.3 m, and T = 11.00 m);
(2) passenger cruisers (L = 260 m, B = 33.0 m, and T = 9.00 m).

Finally, the following values of the safe fairway widths were adopted:

(1) for the straight sections, d = 110 m;
(2) for the transitional sections, d = 130 m (min. 250 m before or after the turn);
(3) for the turns, d = 150 m.

The presented research was supposed to answer if it would be possible to allow the entrance of bigger bulk carriers at the port of Szczecin in those designed fairway parameters. This need was dictated by the plans to call on large bulk carriers at the port of Szczecin and Police, and taking into account the current modernisation of the fairway, it is reasonable to check what the maximum ship is that can actually enter the port. Moreover, at the present stage, it is possible to redesign the fairway so that it will allow the entry of the tested vessel. The final decision will be made by the economic calculations, which unfortunately goes beyond the scope of this publication.

Among all the tests carried out, the most favourable were those using PNS, regardless of the time of day. In Table 6, there is a comparison of the tested bulk carrier (L = 260 m, B = 36.0 m, and T = 11.00 m) fairway widths (good visibility with the PNS system) with designing the fairway widths. In the straight sections, where the assumed width of the fairway is 110 m, differences can be seen to the disadvantages of the designed widths (value given in red colour). In the sections on Zalew Szczeciński, the difference is 10 m, and in the sections limited by the proximity of the shore, the difference is 30 m. However, the sections with bends, due to the fact that they are designed to a width of 150 m, all meet the criteria of a safe fairway width for the tested bulk carrier. It should be noted that the results obtained do not clearly mean that a ship with such parameters cannot enter the port. The calculated widths assume independent passing through the Świnoujście–Szczecin fairway without towing assistance (except for mooring operations). Assuming the deepening of the fairway to 12.5 m, which is currently taking place, and with the use of towing assistance, entering the port of Szczecin is very feasible. This increases the costs of operating the ship, and, as already mentioned, the final decision will be the result of many factors, primarily economic ones.

**Table 6.** Comparison of the tested bulk carrier fairway widths (good visibility with the PNS system) with the designed fairway widths in m.

| Fairway Section | Beginning of Section (km) | Good Visibility (PNS—Daytime) | Fairway Widths |
|---|---|---|---|
| Świnoujście Entrance | −0.5 | 120.00 | 150 |
| Kosa Turn | 2.7 | 146.00 | 150 |
| Mieliński Canal | 5.4 | 120.00 | 150 |
| Mielin Turn | 6.275 | 146.00 | 150 |
| Karsibórz Ferry | 7.55 | 120.00 | 150 |
| Paprotno Turn | 10.2 | 146.00 | 150 |
| Piastowski Canal | 11.1 | 120.00 | 110 |
| Zalew Szczeciński "A" part | 16.8 | 120.00 | 110 |
| Zalew Szczeciński "B" part | 21.9 | 120.00 | 110 |
| Zalew Szczeciński "C" part | 26.45 | 120.00 | 110 |
| Zalew Szczeciński "D" part | 29.85 | 120.00 | 110 |
| Zalew Szczeciński "E" part | 35.5 | 120.00 | 110 |
| Zalew Szczeciński "F" part | 40.1 | 120.00 | 110 |
| Żuławy Turn | 41.36 | 146.00 | 150 |
| Mańków | 42.4 | 120.00 | 150 |
| Szeroki Nurt | 46.3 | 120.00 | 150 |
| Police Turn | 49 | 146.00 | 150 |
| Domiąża | 50.5 | 120.00 | 110 |
| Mnisi Turn | 51.956 | 146.00 | 150 |
| Skolwiński Ostrów | 53.2 | 120.00 | 110 |
| Wyspa Żurawia Turn | 54.6 | 146.00 | 150 |
| Stołczyn | 56 | 120.00 | 110 |
| Radolin Turn | 59.6 | 146.00 | 150 |
| Przekop Mieleński | 64.1 | 120.00 | 110 |
| Parnica Turn | 66.8 | 146.00 | 150 |

## 6. Conclusions

Based on the presented methodology, it was established that the maximum vessel that can navigate the Świnoujście–Szczecin fairway in the worst acceptable conditions after modernisation is a passenger cruiser of the following parameters: L = 260 m, B = 33.0 m, and T = 9.00m and a bulk carrier of the parameters L = 220 m, B = 32.3 m, and T = 11.00 m. It was necessary to check whether the larger bulk carrier had a chance to safely navigate in the above-mentioned fairway. If so, also, under what conditions. The conditions adopted for the analysis were the most favourable possible. Different times of the day, visibility, and different methods of determining the positions were taken into account. As can be seen from the given results, there are no clear conditions under which a bulk carrier with the adopted parameters can safely navigate in the area of the Świnoujście–Szczecin fairway. The PNS system provides sufficient accuracy but only on the straight sections, while at the curves, better results are obtained using the terrestrial method of determining the position of the ship in the fairway. The results obtained do not clearly indicate that it is impossible to enter the port of Szczecin with such a large bulk carrier. Looking only at the draft of the vessel and the available depth, it should be possible, but the poor manoeuvrability of this vessel puts such an operation into question. The use of tugs, as the means to minimise the VTE, may be a solution.

**Author Contributions:** Conceptualisation, A.B.; methodology, P.Z.; investigation, A.B.; resources, A.B. and P.Z.; data curation, A.B.; writing—original draft preparation, A.B.; and writing—review and editing, P.Z. All authors have read and agreed to the published version of the manuscript.

**Funding:** This research received no external funding.

**Institutional Review Board Statement:** Not applicable.

**Informed Consent Statement:** Not applicable.

**Data Availability Statement:** Data available in a publicly accessible repository.

**Conflicts of Interest:** The authors declare no conflict of interest.

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
