# Peer review of "Determination of the Waterway Parameters as a Component of Safety Management System"

_applsci, doi:10.3390/app11104456_

Round 1

Reviewer 1 Report

  • chapter 2.1 : there are a lot of technical errors (repetition of words), also why italic
  • chapter 2.2.: only tables, add text (additional explanation)
  • table 1: abbreviation ''km'' should be after number (not before)
  • table 2: column ''Vessel speed'', what is abbreviation ''w''
  • standardize the notation of a decimal place, somewhere a dot somewhere a comma
  • Symbol for nautical mile: use M or NM or nmi (not Nm)
  • line 268: missing text    (1-?)=0.95
  • Conclusion?

Author Response

Dear Sir or Madam,

Thank you for your effort to review our article. Here are our answers and explanations:

Ad.1 "chapter 2.1 : there are a lot of technical errors (repetition of words), also why italic" - in original all is not in italic, it was because of converting to pdf format. Repetitions of words were corrected.

Ad.2. "chapter 2.2.: only tables, add text (additional explanation)" - corrected.

Ad.3. "table 1: abbreviation ''km'' should be after number (not before)" - corrected.

Ad. 4. "table 2: column ''Vessel speed'', what is abbreviation ''w''" - corrected to "kn - knots".

Ad. 4. "standardize the notation of a decimal place, somewhere a dot somewhere a comma" - corrected to comma as a decimal symbol.

Ad. 5. "Symbol for nautical mile: use M or NM or nmi (not Nm)" - corrected to "NM"

Ad. 6 "line 268: missing text    (1-?)=0.95" - corrected to alpha symbol.

Ad. 7. "Conclusion?" - included.

Reviewer 2 Report

  • The topic of the article is very interesting.
  • Abstract - please provide in the abstract main findings, main goals, not just the way how you achieved the results. It is very unusual to refer in the abstract to references.
  • In the article is missing deep literature review. It is necessary.
  • Chapter 2.1 – Why is this chapter all in italic? In this chapter is missing title. Whole paragraph is unclear written with a lot of duplicates. Moreover, information’s provided in this chapter for sure do not fit in the „Materials and Methods“.
  • Chapter 2.2 – Why is also this sub-chapter in the section „Materials and Methods “?
  • Line 61 – please revise.
  • In whole article is missing numbering of the formulas,
  • In whole article is a lot of typos & missing sings,
  • Line 134 – Please, adjust which figure is correct,
  • Line 164-165, please edit font and size,
  • In the section Materials and Methods there is a lot of formulas which enter into the fairway calculation. But there is need to provide appropriate dataset as a proof of correct calculation.
  • The structure of the article is not appropriate. There is missing conclusion, the discussion section provide only results without any discussion.
  • I cannot recommend this paper for publication. It does not meet basic criteria’s for publishing in scientific journal. Moreover, I must admit that reanalysing the whole work was quite difficult for me.

Author Response

Dear Sir or Madam,

Thank you for your effort to review our article. Here are our answers and explanations:

Ad.1. "Abstract - please provide in the abstract main findings, main goals, not just the way how you achieved the results. It is very unusual to refer in the abstract to references." - corrected.

Ad.2. "In the article is missing deep literature review. It is necessary." - Literature review was added in paragraph 2.

Ad.3. "Chapter 2.1 – Why is this chapter all in italic? In this chapter is missing title. Whole paragraph is unclear written with a lot of duplicates. Moreover, information’s provided in this chapter for sure do not fit in the „Materials and Methods“." - Italic was because of wrong converting to pdf format, in original there is normal text. The chapter was redesigned and corrected according to your remarks.

Ad.4. 'Chapter 2.2 – Why is also this sub-chapter in the section „Materials and Methods “?" - deleted and corrected.

Ad.5. "Line 61 – please revise." - again, wrong converting to pdf file, corrected.

Ad.6. "In whole article is missing numbering of the formulas," - added.

Ad.7. "In whole article is a lot of typos & missing sings," - corrected.

Ad.8. "Line 134 – Please, adjust which figure is correct," - corrected, Figure 4 was the proper one.

Ad.9. - "Line 164-165, please edit font and size," - corrected.

Ad.10. - "In the section Materials and Methods there is a lot of formulas which enter into the fairway calculation. But there is need to provide appropriate dataset as a proof of correct calculation." - included in sub-chapter 4.1 and Table 4. Moreover, thanks to your remarks we have found mistakes in calculation, so the table 5 is changed too. Now is correct.

Ad.11. "The structure of the article is not appropriate. There is missing conclusion, the discussion section provide only results without any discussion." - The structure was corrected, conclusion included.

Ad. 12. "I cannot recommend this paper for publication. It does not meet basic criteria’s for publishing in scientific journal. Moreover, I must admit that reanalysing the whole work was quite difficult for me." - Hoping, the article now meets criteria for publishing.

Reviewer 3 Report

In my opinion this is a worthwhile paper in terms of the practical application and the foundations of the methods involved. The paper is very strong in terms of demonstrating the practical significance of using a tool. To be accepted in the journal I would like to recommend the following corrections:

  1. The literature survey should be expanded
  2. The significance of the method behind the software in comparison to other methods should be well explained.
  3. There has to be more explanation on aspects of traffic management, complexity and the ability of the tool and method to idealise collision and grounding situations. For example refer to : 

Gil, Mateusz; Montewka, Jakub; Krata, Przemyslaw; Hinz, Tomasz; Hirdaris, Spyros (2020). Determination of the dynamic critical maneuvering area in an encounter between two vessels Operation with negligible environmental disruption, Ocean Engineering, 213,  107709.

Zhang, M., Montewka, J., Manderbacka, T., Kujala, P. & Hirdaris, S.(2020)Analysis of the Grounding Avoidance Behavior of a Ro-Pax Ship in the Gulf of Finland using Big Data, 30th International Ocean and Polar Engineering Conference. INTERNATIONAL SOCIETY OF OFFSHORE AND POLAR ENGINEERS, p. 3558-3567 10 p. (Proceedings of the International Offshore and Polar Engineering Conference; vol. 2020-October).

Author Response

Dear Sir or Madam,

Thank you for your effort to review our article. Here are our answers and explanations:

Ad.1. "The literature survey should be expanded" - corrected and included in chapter 2.

Ad.2. The significance of the method behind the software in comparison to other methods should be well explained. - it was explained in paragraph 3 - the method gives results closer to simulation studies due to probabilistic component based on statistical analysis of similar vessels.

Ad.3. "There has to be more explanation on aspects of traffic management, complexity and the ability of the tool and method to idealise collision and grounding situations. For example refer to : " - more explanation added in paragraph 2 and first literature position was added in the text and literature list:

Gil, Mateusz; Montewka, Jakub; Krata, Przemyslaw; Hinz, Tomasz; Hirdaris, Spyros (2020). Determination of the dynamic critical maneuvering area in an encounter between two vessels Operation with negligible environmental disruption, Ocean Engineering, 213,  107709.

Reviewer 4 Report

It seems to be a novel and interesting study. Still, some work needs to be done to bring it to completion.

Consider removing the use of references in the abstract. This is not appropriate.

There are some issues with the formatting in the generated pdf e.g. section 2.1. Pls, take care of that in the revised paper version.

Pls, add a lit review section citing similar research for better contextualisation.

Could you also elaborate in more detail on the need to consider the probabilistic part in your analysis?

Pls, provide numbers to the equations.

Figure 4 has been used in another publication, https://www.sciencedirect.com/science/article/pii/S2092678219300202?via%3Dihub Pls consider removing or getting permission or changing it.

The letters size changes along with the manuscript. Pls, consider amending that.

Line 268 : the confidence level (1-a)?                                  

In this study, from what I understood, only the bulk carrier has been studied. Pls, remove the reference to the other types of vessels in the discussion like passenger vessels (line 292).

The conclusions part is missing.

Minor language polishing is required.

Author Response

Dear Sir or Madam,

Thank you for your effort to review our article. Here are our answers and explanations:

Ad.1. "Consider removing the use of references in the abstract. This is not appropriate." - removed.

Ad.2. "There are some issues with the formatting in the generated pdf e.g. section 2.1. Pls, take care of that in the revised paper version." - corrected.

Ad.3. "Pls, add a lit review section citing similar research for better contextualisation." - added in paragraph 2.

Ad.4. "Could you also elaborate in more detail on the need to consider the probabilistic part in your analysis?" - elaborated in paragraph 2 and expanded in 3.

Ad. 5. 'Pls, provide numbers to the equations." - added.

Ad. 6. "Figure 4 has been used in another publication, https://www.sciencedirect.com/science/article/pii/S2092678219300202?via%3Dihub Pls consider removing or getting permission or changing it." - redrawn.

Ad. 7. "The letters size changes along with the manuscript. Pls, consider amending that." - corrected.

Ad. 8. "Line 268 : the confidence level (1-a)?" - corrected to alpha symbol.                                 

Ad. 9. 'In this study, from what I understood, only the bulk carrier has been studied. Pls, remove the reference to the other types of vessels in the discussion like passenger vessels (line 292)." - that part concerns another research where such vessels were analyzed, so we decided to include them as a whole.

Ad. 10. "The conclusions part is missing." - conclusion included now.

Ad. 11. "Minor language polishing is required." - revised and corrected.

Round 2

Reviewer 2 Report

Dear authors,

The revised version shows an improvement in the quality of the article, its clarity as well as a clear line. I appreciate your effort for revision, and I recommend your article for publishing with minor revisions.

In the literature review, it would be appropriate to look at the problem of safety of navigation in limited conditions also from the point of view of the vessel/ship and not just the waterway. It would be appropriate to mention the methods and research that focuses on the issue of optimizing the shape of the hull, or propulsion. Combination of these two views can be effective in determining the desired waterway parameters.

I recommend including this short approach in the literature review. Below are recommended works that may be helpful to confront in this section.

Buchler, D .; Luck, R .; Markert, M. Propulsion and control system for shallow water ships based on surface cutting double Propellers. In Proceedings of the 8th IFAC Conference on Control Applications in Marine Systems, Rostock-Warnemunde, Germany, 15–17 September 2010.

Illes, L .; Kalina, T .; Jurkovic, M .; Luptak, V. Distributed Propulsion Systems for Shallow Draft Vessels. J. Mar. Sci. Eng. 2020, 8, 667. https://doi.org/10.3390/jmse8090667

Raven, H. A new correction procedure for shallow-water effects in ship speed trials. In Proceedings of the 13th International Symposium on PRActical Design of Ships and Other Floating Structures PRADS ’2016, Copenhagen, Denmark, 4–8 September 2016.

Rotteveel, E .; Hekkenberg, R .; Ploeg, A. Inland ship stern optimization in shallow water. Ocean. Eng. 2017, 141, 555–569, doi: 10.1016 / j.oceaneng.2017.06.028.

Author Response

Good day,

Thank you for your effort to revise our paper. The literature you mentioned was included in our paper together with short comment in section 2 "Related work".

Best regards,

Reviewer 3 Report

The paper scientific basis is improved and herefore in my opinion can be accepted. Please add references :

Remote piloting in an intelligent fairway – A paradigm for future pilotage - ScienceDirect

A Big Data Analytics Method for the Evaluation of Ship - Ship Collision Risk reflecting Hydrometeorological Conditions. https://doi.org/10.1016/j.ress.2021.107674 https://www.sciencedirect.com/science/article/pii/S095183202100212X

Author Response

(The authors gave the same response as above.)

Reviewer 4 Report

I can see with satisfaction that my comments have been sufficiently addressed. I would recommend that authors implement a polishing of the language prior to the final submission to improve the quality of the publication.

Author Response

Good day,

Thank you for your effort to revise our paper. The language was revised by English teacher, hope now is better.

Best regards,